# Effects of Anisotropy on Single Crystal Silicon in Polishing Non-Continuous Surface

**DOI:** 10.3390/mi11080742

**Published:** 2020-07-30

**Authors:** Guilian Wang, Zhijian Feng, Yahui Hu, Jie Liu, Qingchun Zheng

**Affiliations:** 1Tianjin Key Laboratory for Advanced Mechatronic System Design and Intelligent Control, School of Mechanical Engineering, Tianjin University of Technology, Tianjin 300384, China; wangguilian@tjut.edu.cn (G.W.); axdfengzhijian@163.com (Z.F.); huyahui@tjut.edu.cn (Y.H.); liujiejane@163.com (J.L.); 2National Demonstration Center for Experimental Mechanical and Electrical Engineering Education, Tianjin University of Technology, Tianjin 300384, China

**Keywords:** single crystal silicon, anisotropy, polishing, non-continuous surface

## Abstract

A molecular dynamics model of the diamond abrasive polishing the single crystal silicon is established. Crystal surfaces of the single crystal silicon in the Y-direction are (010), (011), and (111) surfaces, respectively. The effects of crystallographic orientations on polishing the non-continuous single crystal silicon surfaces are discussed from the aspects of surface morphology, displacement, polishing force, and phase transformation. The simulation results show that the Si(010) surface accumulates chips more easily than Si(011) and Si(111) surfaces. Si(010) and Si(011) workpieces are deformed in the entire pore walls on the entry areas of pores, while the Si(111) workpiece is a local large deformation on entry areas of the pores. Comparing the recovery value of the displacement in different workpieces, it can be seen that the elastic deformation of the A side in the Si(011) workpiece is larger than that of the A side in other workpieces. Pores cause the tangential force and normal force to fluctuate. The fluctuation range of the tangential force is small, and the fluctuation range of the normal force is large. Crystallographic orientations mainly affect the position where the tangential force reaches the maximum and minimum values and the magnitude of the decrease in the tangential force near the pores. The position of the normal force reaching the maximum and minimum values near the pores is basically the same, and different crystallographic orientations have no obvious effect on the drop of the normal force, except for a slight fluctuation in the value. The high-pressure phase transformation is the main way to change the crystal structure. The Si(111) surface is the cleavage surface of single crystal silicon, and the total number of main phase transformation atoms on the Si(111) surface is the largest among the three types of workpieces. In addition, the phase transformation in Si(010) and Si(011) workpieces extends to the bottom of pores, and the Si(111) workpiece does not extend to the bottom of pores.

## 1. Introduction

The single crystal silicon has a basically complete lattice structure with different properties in different directions. In order to better understand the mechanical and chemical properties of single crystal silicon, it is necessary to fully understand their properties in each crystallographic orientation. Previous studies have shown that the influence of anisotropy on the processing of materials is mainly reflected in the accumulation of materials, elastoplastic properties, and phase transformation during the processing [1,2,3,4], which often play a decisive role in the quality of materials. For porous crystalline materials, the effect of crystallographic orientations on machining is more complicated. Because of the special geometric structure, the bearing capacity of porous materials will be reduced, and it is difficult to obtain an ideal surface using traditional processing methods [5,6,7,8,9]. Therefore, grasping the structural and property differences caused by the anisotropy of the porous crystalline material during the processing is crucial for selecting the process parameters to obtain an ideal processing surface.

Different crystallographic orientations bring various structures and peculiar mechanical properties, which widen the application of crystal materials. However, it also causes many problems in processing and manufacturing. In order to obtain crystals with higher precision and better mechanical properties, many scholars have done extensive research on the anisotropy of crystals. Filippov et al. [10] studied the influence of crystallographic orientations and azimuth indenter orientations on indentation hardness and modulus by Vickers indentation on (100), (110), and (111) surfaces of single crystal aluminum. Tang et al. [11] compared the anisotropic behavior of crystals with experimental results and found that the extended anisotropic Drucker yield criterion can accurately simulate the anisotropy of FCC (face-centered cubic), BCC (body-centered cubic), and HCP (hexagonal close-packed) metals. Armstrong et al. [12] conducted stress calculations on the (111) and (100) surfaces of α-iron, and discovered the dislocation reaction mechanism of enhanced strain hardening during the process of the nano-indentation. Kitsuya et al. [13] studied crystallographic orientations on some metals, and the crystal plasticity analysis based on the information obtained from experiments showed that the anisotropy of mechanical properties is greatly affected by the crystallographic orientations. Bürger et al. [14] tested the shear creep behavior of nickel-based single crystal superalloys and found that the shear deformation in the [112] loading direction is much faster than that in the [011] loading direction on the (110) surface. Zhan et al. [15] reported the mechanical deformation of the single crystal Ti2AlC MAX phase using microcolumn compression tests with a series of crystallographic orientations, and found that non-classical crystallographic slip is caused by the strong dependence of crystallographic slip on both the analytical shear stress and the stress perpendicular to the slip plane. Frydrych et al. [16] studied the yield stress, hardened texture, and twin anisotropy of AZ31B extruded bars, and concluded that the twinning activity mainly affects the mechanical response through texture changes. Xie et al. [17] studied the removal mechanism of single crystal copper in the close-to-atomic scale by molecular dynamics, and simulated the influence of anisotropy on cutting force and material distribution of the surface. It was found that the minimum chip thickness can be reduced to a single atomic layer in an atomic scale using rounding tools for mechanical cutting. Frodal et al. [18] studied three aluminum alloys with different grain structures and crystal textures, and found that effects of yield strength and work hardening depend on plastic anisotropy. Pogrebnjak et al. [19,20] studied two kinds of nano-multilayer films CrN/MoN and TiN/SiC through experiments, and found that the preferred crystallographic orientation will change under different voltages. Meanwhile, the effect of temperature on the physicochemical and functional properties of the TiN/SiC multilayer film was explored. The research results are consistent with the first principle MD (molecular dynamics) calculation of the TiN/SiC heterostructure. The above studies on metals and alloys show that the anisotropy of crystals will have a significant impact on the yield strength, work hardening, and shear slip dislocation of metals and alloys. Reasonable selection of crystallographic orientations can effectively optimize the processing of metals and alloys, and improve the processing quality.

Unlike metals and alloys, some studies have found that hard and brittle crystal materials are prone to cracks and chipping during the processing, and the crystallographic orientation is closely related to the generation of cracks during the processing of brittle materials. In order to overcome the influences of the anisotropy on the processing of hard and brittle materials, it is imperative to study the anisotropy of hard and brittle materials. Meng et al. [21] focused on the influences of crystallographic anisotropy on 6H-SiC slip deformation and nanomachining performance, and made important contributions to understand the microdeformation and nanomachining processes of 6H-SiC. Cang et al. [22] observed obvious elastic anisotropy in glass and found that the elastic anisotropy shows a strong correlation with molecular orientations. Chen et al. [23] did some research on cutting SiO2 and found that (100) [00-1] crystallographic orientation has a large range of damage extension, (110) [1–10] crystallographic orientation has the smallest damage range, and a new phase is generated in the (111) [-101] crystallographic orientation. Nagai et al. [24] discussed the formation mechanism of U-shaped diamond grooves based on the experimental results, and successfully obtained a U-shaped diamond trench with vertical {111} sidewalls for power devices. Liu et al. [25] analyzed the internal reasons for the anisotropy of the frictional force when the diamond slided on the diamond, which was mainly caused by amorphous defects, dislocations, and von Mises shear strain. Zhao et al. [26] simulated the crack fracture modes of the glass layer and silicon layer with (100), (110), and (111) surfaces. The changes of crack edge quality in different cutting directions and different layers were obtained. It was found that anisotropy of the silicon layer has an important influence on the fracture mode and crack edge quality of the two layers. Rickhey et al. [27] analyzed changes of the crack size in Si(001), Si(110), and Si(111) orientations, which were in good agreement with experiments and proved the rationality of the simplified numerical model. These studies about the anisotropy of hard and brittle crystalline materials provide some good theoretical and experimental references for the evolution of material cracks during the processing. In addition, the change in crystal orientations also has a significant effect on the friction and elasticity between brittle materials.

Although these researchers have conducted many studies on the anisotropy of non-porous crystals, it is still necessary to describe the anisotropy of porous crystalline materials in detail. In this paper, pores make the single crystal silicon workpiece become a non-continuous geometry, which has some properties that the continuous surface does not have. The influence of anisotropy on the processing of porous materials is very easy to change to some extent owing to the heterogeneity of the materials. In order to explore how the anisotropy affects the pore walls, a molecular dynamics model is established in this paper, which is about polishing the non-continuous surface. Laws about the removal, stress, and modification of porous materials are revealed by changing crystallographic orientations. Meanwhile, studies on the distribution of the materials, displacement, polishing force, and phase transformation are conducted. These studies are very important for exploring the effect of crystallographic orientations on porous materials and studying the removal mechanism of porous materials.

## 2. Simulation Method

In this paper, a set of three-dimensional molecular dynamics simulation models are established. The schematic diagram of the models is shown in Figure 1.

The model is mainly composed of the diamond abrasive and the single crystal silicon workpiece. The diamond abrasive is a sphere with a radius of 2 nm. The workpiece is rectangular with two pores, whose dimensions in the X-direction, Y-direction, and Z-direction are 17 nm, 11 nm, and 11 nm, respectively. Pores are positioned at the polishing distance of 4–6 nm and 8–10 nm, and the sizes in the X-direction, Y-direction, and Z-direction are 2 nm, 3 nm, and 6 nm, respectively. For the convenience of descriptions, pore walls of the workpiece are marked in Figure 1b. A and C sides are pore walls in the entry areas of pores. B and D sides are pore walls in the exit areas of pores. Three kinds of crystal surfaces are set in the model, namely, (010), (011), and (111) surfaces. The structure of the crystal surface and the atomic spacing ‘d’ of single crystal silicon are shown in Figure 2. The maximum value of the atomic spacing ‘d’ determines properties of the crystal. Figure 3 shows differences in the shape of the first pore under different crystallographic orientations. In this paper, the polishing depth is expressed by ‘ap’.

The workpiece is divided into three parts: Newtonian atoms, thermostatic atoms, and boundary atoms. Newtonian atoms are atoms in the polishing area, thermostatic atoms keep the temperature constant, and boundary atoms are fixed. The microcanonical ensemble (NVE) is used, the initial temperature of the simulation is kept in 293 K, and periodic boundary conditions (PBC) are set in the Z-direction. The abrasive is set as a rigid body, the interaction between C-C atoms is negligible, the interaction between C-Si atoms is described by Morse potential function [28], and the Morse potential function can be expressed by the following equation:(1)$$E(r_{ij})=D_{0}[e^{-2\alpha (r_{i}{}_{j}-r_{0})}-2e^{-2\alpha (r_{ij}-r_{0})}]$$
where $D_{0}$ is the binding energy, α is the gradient coefficient of the potential energy curve, and $r_{0}$ is the distance between the two atoms when the interaction between atoms is equal to zero.

The interaction between Si-Si atoms is described by the Tersoff potential function [29], and the Tersoff potential function can be expressed by the following equations:(2)$$E=\frac{1}{2}\sum_{i\neq j}U_{ij}$$
(3)$$U_{ij}=f_{c}(r_{ij})[f_{R}(r_{ij})+b_{ij}f_{A}(r_{ij})]$$
where $f_{c}(r_{ij})$ is the cut-off function, $f_{R}(r_{ij})$ is the rejection function, $f_{A}(r_{ij})$ is the attraction function, $b_{ij}$ is the modulation function, and $r_{ij}$ is the distance between atomic *i* and *j*. $f_{R}(r_{ij})$, $f_{A}(r_{ij})$ and $f_{c}(r_{ij})$ can be expressed by the following equations:(4)$$f_{R}(r_{ij})=A_{ij}e^{-\lambda_{ij}r_{ij}}$$
(5)$$f_{A}(r_{ij})=-B_{ij}e^{-\mu_{ij}r_{ij}}$$
(6)$$f_{c}(r_{ij})=\left\{ \begin{array}{l} 1\\ \frac{1}{2}+\frac{1}{2}\cos(\frac{r_{ij}-R_{ij}}{S_{ij}-R_{ij}})\\ 0 \end{array}\right.\begin{array}{l} r_{ij}\leq R_{ij}\\ R_{ij}<r_{ij}\leq S_{ij}\\ r_{ij}>S_{ij} \end{array}$$
where $A_{ij}$ is dual binding energy of the rejection function, $B_{ij}$ is dual binding energy of the attraction function, $S_{ij}$ and $R_{ij}$ are cut-off radii, $\lambda_{ij}$ is the gradient coefficient of the potential energy curve in the attraction function, and $\mu_{ij}$ is the gradient coefficient of the potential energy curve in the rejection function.

$b_{ij}$ can be expressed by the following equations:(7)$$b_{ij}=x_{ij}{\left(1+\beta_{i}^{n_{i}}\zeta_{ij}^{n_{i}}\right)}^{-\frac{1}{2n_{i}}}$$
(8)$$\zeta_{ij}=\sum_{k\neq i,j}f_{c}(r_{ik})\omega_{ik}g(\theta_{ijk})$$
(9)$$g(\theta_{ijk})=1+\frac{c_{i}^{2}}{d_{i}^{2}}-\frac{c_{i}^{2}}{d_{i}^{2}+(h_{i}-\cos\theta_{ijk})}$$
where $\beta_{i}$ is the bond order coefficient; $c_{i}$, $d_{i}$, and $h_{i}$ are elastic coefficients; $\zeta_{ij}$ is angular potential energy; and $\theta_{ijk}$ is the angle of atoms.

The simulation is divided into two parts: system relaxation and the polishing process of the workpiece. The relaxation process is mainly to keep the system energy stable and make the simulation environment close to the actual situation. During the polishing process, the abrasive feeds in the negative direction of the X-axis and rotates with the Z-axis. All MD simulations are based on the large-scale atomic/molecular massively parallel simulator (LAMMPS) [30] and results are visualized by OVITO [31]. Curves are drawn by ORIGIN [32].

Simulation parameters are shown in Table 1.

## 3. Results and Discussion

### 3.1. Analysis of the Surface Topography and Displacement Vector 

#### 3.1.1. Analysis of the Surface Topography

The anisotropy of the crystal is a key parameter that determines the structure and performance of the material, which affects the removal method of materials and has an important impact on the processing efficiency and quality. Figure 4 shows three kinds of single crystal silicon workpieces with Si(010), Si(011), and Si(111) surfaces in the Y-direction, and describes the distribution of materials on the surfaces at the end of polishing. It is obvious that the removed materials mainly exist in the following ways: covering the surface of pores, accumulating at the front of the abrasive to be chips, and some of the chips being converted to the side flow. Figure 4d describes the conversion process of chips and side flows. When the polishing depth is 0.75 nm, the coverage of pore areas is low. A few chips and side flows are formed. When the polishing depth is 1.5 nm, the coverage of pore areas is very high, and a large amount of chips and side flows are formed. Some reasons can account for these phenomena. When the polishing depth becomes larger, the contact part between the abrasive and the workpiece is improved, and the polishing area is widened, which improves the coverage of materials in pore areas. Meanwhile, the amount of materials removed is increased with increase in the polishing depth, so the increase of chips is more obvious, and there are more chips converted into the side flows. Generally speaking, side flow atoms are less than chip atoms. Comparing the distribution of materials on the three kinds of surfaces at the same polishing depth, it is easy to find that the number of chips gathered on the Si(010) surface is more than that on Si(011) and Si(111) surfaces at the end of polishing, and the number of chips produced on the Si(111) surface is the least, which indicates that the resistance of the unpolished surface to the abrasive is smaller and materials accumulate more easily when polishing the Si(010) surface. In contrast, the Si(111) surface will lead to the less accumulation of materials during the polishing.

#### 3.1.2. Effects of Anisotropy on the Deformation of Pores

The deformation process of each pore wall (A, B, C, and D sides) is not identical. Generally speaking, the deformation of A and C sides is basically the same, and the deformation of B and D sides is basically the same. In the numerical simulation, the movement direction of A and C sides points to the pore areas, and the movement direction of B and D sides points to the continuous surface, which indicates that the deformation of pore walls is related to the relative movement direction of the abrasive [33]. In order to study the effects of anisotropy on the deformation of pore walls more reasonably, the deformation of the first pore in different crystallographic orientations is analyzed. In Figure 5 and Figure 6, the length of blue arrows represents the size of the displacement, and the direction of arrows represents the displacement direction of atoms relative to the initial position. Figure 5a–c describes the deformation of the A side in different crystallographic orientations when the polishing distance is 4 nm. While Figure 6a–c describes the displacement of atoms at the first pore when the polishing distance is 14.2 nm. It can be seen from Figure 5 that, when the surface in the Y-direction is (111) surface, the displacement of atoms in the upper part of the A side is relatively large. Blue arrows are relatively dense and long. The bending deformation with a large angle is obvious in the upper part of the A side, but the deformation in the lower part of the A side is not obvious, which is the local deformation. When the surface in the Y-direction is (010) surface and (011) surface, the deformation of the A side belongs to the entire deformation. When the surface in the Y-direction is (011) surface, the deformation of the A side is the smallest, and the bending angle is the smallest. It can be seen from the Figure 6 that, when the polishing distance is 14.2 nm, the displacement of the upper part of the first pore changes with the change of the surface in the Y-direction. When the surface in the Y-direction is (111) surface, the number of blue arrows is large and the atomic displacement near the pore wall is large as well. This indicates that, when the surface of the workpiece in the Y-direction is (111) surface, the deformation of pore walls is large, and the atoms above the pore are prone to the large displacement. This is mainly because the Si(111) surface is the cleavage surface of single crystal silicon. The distance parallel to the atomic layer in the cleavage surface is relatively large, which easily contributes the atomic layer having cracks owing to the extrusion pressure, so the pore wall easily produces large deformation, and the atoms nearby are prone to the large displacement.

In order to study numerical characteristics of the atomic displacement in pore walls more intuitively, the pore wall is sliced and the average displacement is extracted. The size of the slice is 0.5×3×4 nm^3^ (0.5 nm in the X-direction, 3 nm in the Y-direction, and 4 nm in the Z-direction). Because the deformation laws of A and C sides are the same, and the deformation laws of B and D sides are the same, this paper only discusses the numerical changes of A and B sides. Figure 7 shows the average displacement of A and B sides under different crystallographic orientations. The average displacement of the A side increases first and then decreases. The average displacement of the B side increases and then gradually flattens. This shows that there is a recovery process in the deformation of the A side, and it can be known that the deformation mechanism of the A side is elastic–plastic mixed deformation. There is no recovery process in the B side, so it can be known that the deformation mechanism of the B side is plastic deformation. In the same way, the deformation mechanism of the C side is elastic–plastic mixed deformation, and that of the D side is plastic deformation. As is shown in Figure 7a, when the polishing distance is 4 nm, the average displacements of the A side in the Si(010), Si(011), and Si(111) workpieces are 0.76 nm, 0.68 nm, and 0.76 nm, respectively. When the polishing distance is 4 nm, the deformation of the A side under different crystallographic orientations is the same as that shown in Figure 5. It is easy to find that the average displacement of the A side in the Si(111) workpiece is always greater than that in other workpieces as the polishing continues. Although the average displacement of the A side in the Si(010) workpiece is slightly higher than that in the Si(011) workpiece at the polishing distance of 4 nm. As the polishing continues, the average displacement of the A side in the Si(010) workpiece is lower than that in other workpieces. The average displacement of the A side in the three kinds of workpieces basically changes with the same trend, which reaches the maximum value at the polishing distance of 6 nm. When the polishing distance is 6 nm, the average displacements of the A side in Si(111), Si(011), and Si(010) workpieces are 1.01 nm, 0.9 nm, and 0.93 nm, respectively. The average displacements of the A side are 0.83 nm, 0.58 nm, and 0.66 nm, respectively, at the polishing distance of 14.2 nm. The recovery values of the displacement are 0.18 nm, 0.32 nm, and 0.27 nm, respectively, which indicates that the Si(011) workpiece has the largest elastic recovery value after the polishing, that is, the elastic deformation of the A side in the Si(011) workpiece is larger than that of other workpieces. The average displacement of the B side in the three kinds of workpieces basically changes with the same trend. The displacement increases obviously in the polishing distance from 6 nm to 10 nm, but it increases slowly after the polishing distance of 10 nm. When the polishing distance is 10 nm, the average displacements of the B side in the Si(111), Si(011), and Si(010) workpieces are 1.0 nm, 0.85 nm, and 0.86 nm, respectively. At the end of polishing, the average displacements of the B side in the Si(111), Si(011), and Si(010) workpieces are 1.22 nm, 1.03 nm, and 1.01 nm, respectively. Generally speaking, the average displacement of the B side in the Si(111) workpiece is always greater than that in the other two workpieces.

### 3.2. Analysis of the Polishing Force

#### 3.2.1. Calculation Method of the Polishing Force

The polishing force directly reflects the material removal process and is an important parameter to understand the processing process. The polishing force mentioned in this paper refers to the real-time average force of the workpiece on the abrasive during the polishing process. In general, the polishing force in the macro scale comes from the deformation of the workpiece and the friction between the tool and chips, while in the micro scale, the polishing force mainly comes from the interaction between workpiece atoms and tool atoms. This section mainly studies the relationship between the normal polishing force (Y-direction) and the tangential polishing force (X-direction) with the polishing distance, aiming to explain the influence of different crystallographic orientations on the polishing force of non-continuous surfaces. The tangential force is recorded as Ft, and the normal force is recorded as Fn.

#### 3.2.2. Changes of the Polishing Force 

Figure 8 shows the change of the polishing force (tangential force and normal force) in three different crystallographic orientations when the polishing depth is 1 nm. It is obvious that there are differences in the polishing force under different crystallographic orientations. However, in terms of the overall trend, the polishing force in the three kinds of crystallographic orientations basically changes with the same trend, so this paper selects Si(010) surface in the Y-direction to describe the effect of pores on the polishing force, as shown in Figure 8a. It can be seen from Figure 8a that the position of pores is at the polishing distance of 4–6 nm and 8–10 nm, but the position where the polishing force reaches the maximum and minimum values does not coincide with the boundary of the pores. In the vicinity of the first pore, the maximum value of the tangential force appears at the polishing distance of 3 nm, the minimum value appears at the polishing distance of 5.2 nm, and the tangential force decreases about 18 nN. In the vicinity of the second pore, the maximum value of the tangential force appears at the polishing distance of 7 nm, the minimum value appears at the polishing distance of 9 nm, and the tangential force decreases about 29 nN. Generally speaking, the fluctuation of the tangential force is relatively small. This is mainly because the source of the tangential force is mainly divided into two parts: one is the reaction force acting on the abrasive when atoms of the workpiece are sheared and the other is the movement of chips in front of the abrasive, which is an important component of the tangential force. For non-continuous surfaces, although the tangential force tends to decrease owing to the lack of atoms in pore areas, the fluctuation amplitude will be relatively weak owing to the existence of chips in front of the abrasive.

There is a big difference between the normal force and the tangential force. In pore areas, the fluctuation range of the normal force is larger than that of the tangential force. In the vicinity of the first pore, the maximum value of the normal force appears at the polishing distance of 3 nm, the minimum value appears at the polishing distance of 6.3 nm, and the normal force decreases about 100 nN. In the vicinity of the second pore, the maximum value of the normal force appears at the polishing distance of 8.3 nm, the minimum value appears at the polishing distance of 10.2 nm, and the normal force decreases about 70 nN. This is because the normal force is provided by the normal reaction force when the workpiece is extruded. The lack of atoms in the pore areas reduces the normal reaction force provided by the workpiece, resulting in the sudden drop of the normal force in the pore areas. By comparing the wave amplitude of the normal force and tangential force, it can be concluded that the wave amplitude of normal force is much larger than that of the tangential force, which shows that the effect of pores on the normal force is larger than that of the tangential force.

It can be seen from Figure 8a–c that the change of crystallographic orientations will bring some local changes to the polishing force. For the tangential force, crystallographic orientations mainly affect the position where the tangential force reaches the maximum and minimum values and the decrease of the tangential force in the pore areas. In the vicinity of the first pore, the maximum value of the tangential force on the Si(011) surface appears at the polishing distance of 4.5 nm, the minimum value appears at the polishing distance of 5.3 nm, and the tangential force decreases about 30 nN. The maximum value of the tangential force on the Si(111) surface appears at the polishing distance of 3.2 nm, the minimum value appears at the polishing distance of 5.8 nm, and the tangential force decreases about 40 nN. In the vicinity of the second pore, the maximum value of the tangential force on the Si(011) surface appears at the polishing distance of 6.8 nm, the minimum value appears at the polishing distance of 9.5 nm, and the tangential force decreases about 25 nN. The maximum value of the tangential force on the Si(111) surface appears at the polishing distance of 7.5 nm, the minimum value appears at the polishing distance of 9.2 nm, and the tangential force decreases about 27 nN. The decrease of the tangential force caused by different orientations is mainly related to the accumulation of chips in front of the abrasive. It can be seen from Figure 4 that chips on the Si(010) surface accumulate easily, and the tangential reaction force given by chips is relatively large, so the tangential force near the first pore on the Si(010) surface decreases the least. However, there is less chips accumulation on the Si(111) surface, and the tangential reaction force given by chips is relatively small, so the tangential force near the first pore on the Si(111) surface decreases the most. Because of the small distance between the first pore and the second pore, the effect of chips on tangential force near the second pore is not obvious, and the difference between the tangential force of different workpieces is only 4 nN.

For the normal force, the position where the normal force reaches the maximum and the minimum values near the pores is basically same. In the vicinity of the first pore, the normal force of the Si(011) workpiece decreases about 100 nN, and the normal force of the Si(111) workpiece decreases about 105 nN. In the vicinity of the second pore, the normal force of the Si(011) workpiece decreases about 57 nN, and the normal force of the Si(111) workpiece decreases about 60 nN. It can be seen that different crystallographic orientations have no obvious effect on the drop of normal force near the first pore, and the difference in the decrease of the normal force near the second pore under different crystallographic orientations is only 13 nN. Those phenomena indicate that the drop of the normal force is mainly affected by the geometry of the workpiece. Besides, when the normal force reaches the peak value near the first pore at the first time, the peak value of the normal force on the Si(010) and Si(011) surfaces is greater than 100 nN, while the peak value of the normal force on the Si(111) surface is lower than 100 nN, which is mainly because (111) surface is the cleavage surface of single crystal silicon, the atomic layer bonding force parallel to the cleavage surface is weak, and the extrusion pressure required for extrusion failure is relatively small.

### 3.3. Phase Transformation Analysis

#### 3.3.1. Principle of Phase Transformation 

Phase transformation analysis is a common method to analyze crystal structure. In this paper, the coordination number is used to study phase transformation. At the beginning, the coordination number of single crystal silicon is 4, and there are four neighboring atoms. When the pressure of the abrasive on the workpiece reaches the critical value, the crystal structure with coordination number 5 and 6 will appear. Figure 9 shows the cross-section of the Si(010) workpiece at the polishing depth of 1 nm. It can be seen from Figure 9 that there will be a large number of atoms under the abrasive, but phase transformation atoms are in an unstable state. When the abrasive leaves, the β-Si loses its crystal order and becomes an amorphous phase. The amorphous phase is mainly composed of disordered covalent bond and a small amount of β-Si. The whole transformation process follows the high pressure transformation principle. ‘CN’ in the Figure 9 represents the coordination number.

#### 3.3.2. Effects of Anisotropy on Phase Transformation

Figure 10 shows the analysis of the main phase transformation atoms in different crystallographic orientations. In order to increase the reliability of the research results, this section conducts comparative simulations by changing the polishing depths (0.75 nm, 1.0 nm, 1.25 nm, 1.5 nm). As can be seen from Figure 10a, for the amorphous atoms with a coordination number of 5, when the polishing depth is 0.75 nm, 1.0 nm, 1.25 nm, and 1.5 nm, the number of phase transformation atoms with coordination number of 5 on the Si(111) surface is 985, 1167, 1656 and 2023, respectively. It can be seen that the number of atoms with the coordination number of 5 on the Si(111) surface is the most at all polishing depths. The number of atoms with the coordination number of 5 on the Si(011) surface is slightly more than that on the Si(010) surface in most conditions, or maintains a small difference. Figure 10b shows that the number of atoms with the coordination number of 6 is far less than that with the coordination number of 5 at the same polishing depth, and the effect of crystallographic orientations on it is similar to that of crystallographic orientations on atoms with coordination number 5. However, when the polishing depth is 0.75 nm, the number of atoms with the coordination number of 6 on the Si(111) surface is the least among the three kinds of workpieces, which is only 208. The number of atoms with the coordination number of 6 on the Si(010) and Si(011) surfaces is 217 and 221, respectively. This shows that the number of atoms with the coordination number of 6 on the Si(111) surface does not increase obviously when the polishing depth is small. Only when the polishing depth reaches a certain degree, the number of atoms with the coordination number of 6 on the Si(111) surface is larger than that on other surfaces. Besides, it can be seen from the Figure 10c that the total number of main phase transformation atoms on the Si(111) surface is the largest among the three types of crystal surfaces and the total number of main phase transformation atoms on the Si(010) surface is the smallest. The total number of main phase transformation atoms on the Si(011) surface is slightly more than that on the Si(010) surface. This is mainly because the Si(111) surface is the cleavage surface of single crystal silicon, which is modified more easily than other surfaces. The Si(011) surface also has weak cleavage owing to its large atomic layer spacing in the Y-direction, but its cleavage is smaller than Si(111) surface.

Figure 11 shows the phase transformation under different crystallographic orientations at the polishing depth of 1.0 nm. From Figure 11a–c, it can be seen that the phase transformation at the A and C sides is always greater than that at B and D sides. Figure 11a,b shows the phase transformation at the pore walls of Si(010) and Si(011) workpieces. It is easy to find that there is amorphous phase transformation extending to the bottom at A and C sides. Figure 11c shows that the phase transformation at A and C sides of the Si(111) workpiece is basically concentrated in the upper part of the pore walls, and does not extend to the bottom. However, the number of phase transformation atoms at A and C sides in the Si(111) workpiece is more than that in Si(010) and Si(011) workpieces. This is basically consistent with the above analysis results of the displacement vector at pore walls; that is, the entire deterioration of A and C sides occurs in the Si(010) and Si(011) workpieces, while the deterioration of A and C sides occurs locally in the Si(111) workpiece, but the deterioration is relatively serious. The main reason for the differences of phase transformation between A and C sides is that the atomic layer spacing on the Si(111) surface in the Y-direction is too large, and the binding force between atoms is relatively weak. The atomic layer spacing on the Si(010) and Si(011) surfaces in the Y-direction is smaller than that on the Si(111) surface, and the binding force between atoms is relatively strong. The atomic bond does not break easily during extrusion, so the entire deformation occurs. When the pressure reaches the critical value, the whole pore wall will have phase transformation, but the scale of phase transformation is relatively small.

## 4. Summary and Conclusions

In this paper, the molecular dynamics model of the single crystal silicon with non-continuous surface is studied. Influences of different crystallographic orientations on polishing the non-continuous surface are explored, and the following conclusions can be drawn:(1)It can be found that, when the surface in the Y-direction is (010) and (011) surfaces, the bending deformation of the A side belongs to the entire deformation. When the surface in the Y-direction is (111) surface, the deformation of the A side belongs to the large local deformation. It is also found that the elastic deformation of A side in the Si(011) workpiece is larger than that in other workpieces. The average displacement of the B side in the Si(111) workpiece is always greater than the other two workpieces. The fluctuation of the tangential force is relatively small and that of the normal force is relatively large. For the tangential force, crystallographic orientations mainly affect the position where the tangential force reaches the maximum and minimum values and the decrease of the tangential force in pore areas. For the normal force, the position of the normal force reaching the maximum and minimum values near the pores is basically the same, and different crystallographic orientations have no obvious effect on the drop of the normal force, except for a slight fluctuation in the value.(2)For the amorphous atoms with the coordination number of 5, atoms with the coordination number of 5 on the Si(111) surface are the most at all polishing depths. For the atoms with the coordination number of 6, effects of crystallographic orientations on it are similar to that on the atoms with the coordination number of 5. When the polishing depth is 1 nm, A and C sides in the Si(010) and Si(011) workpieces show overall deterioration, and the transformation extends to the bottom of the pores. A and C sides in the Si(111) workpiece show local deterioration, which is serious, but does not extend to the bottom of the pores.

## Figures and Tables

**Figure 1 micromachines-11-00742-f001:**
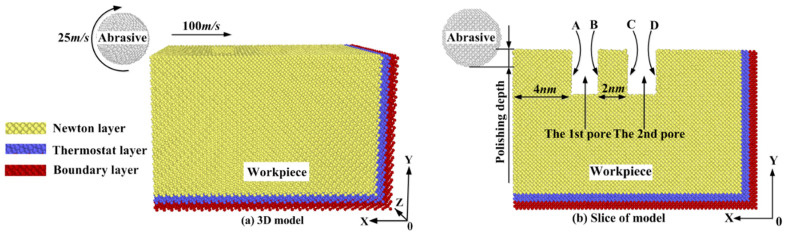
Schematic diagram of the diamond abrasive polishing single crystal silicon.

**Figure 2 micromachines-11-00742-f002:**
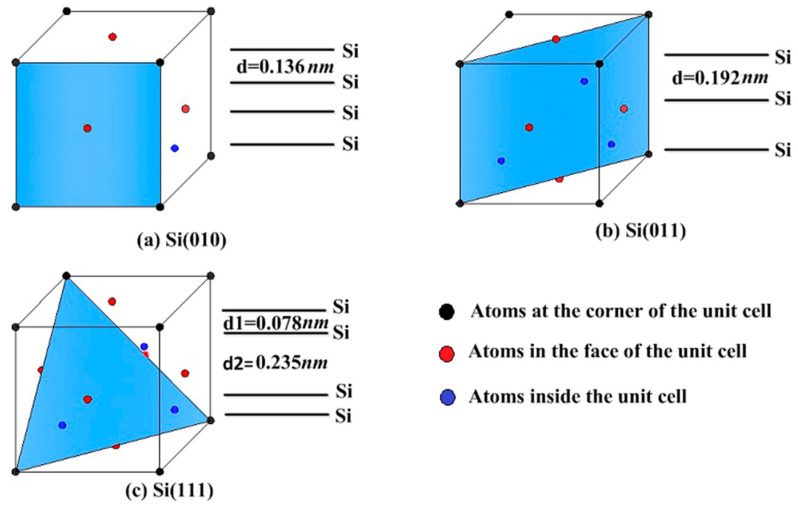
Structure and atomic spacing of single crystal silicon.

**Figure 3 micromachines-11-00742-f003:**
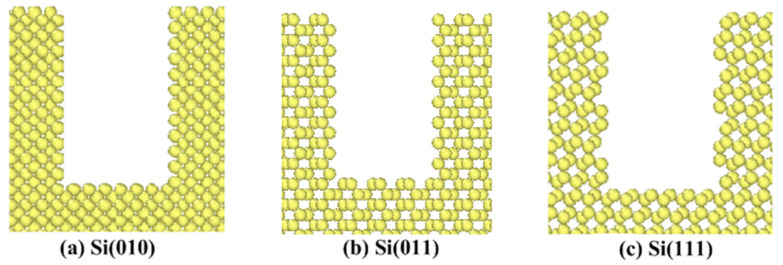
Influences of the anisotropy on pore walls.

**Figure 4 micromachines-11-00742-f004:**
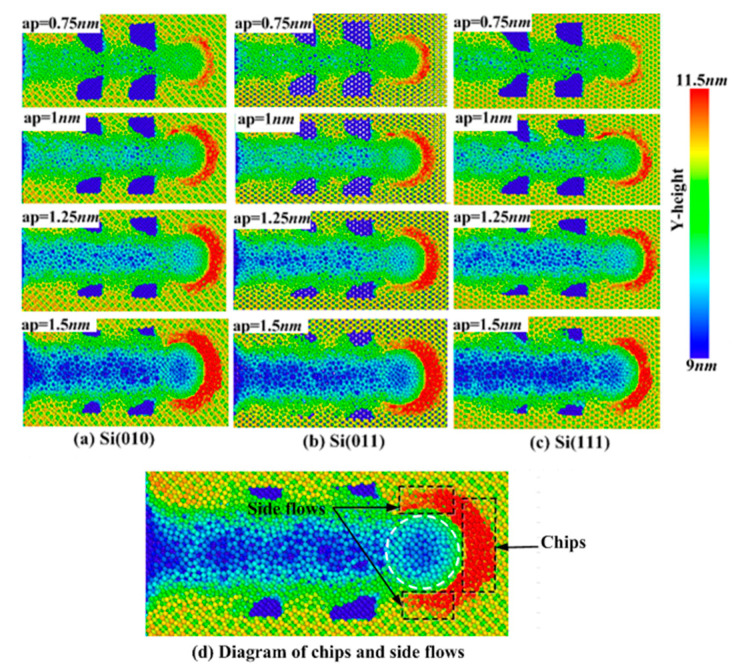
Effects of anisotropy on the surface morphology at different polishing depths (0.75 nm, 1 nm, 1.25 nm, 1.5 nm).

**Figure 5 micromachines-11-00742-f005:**
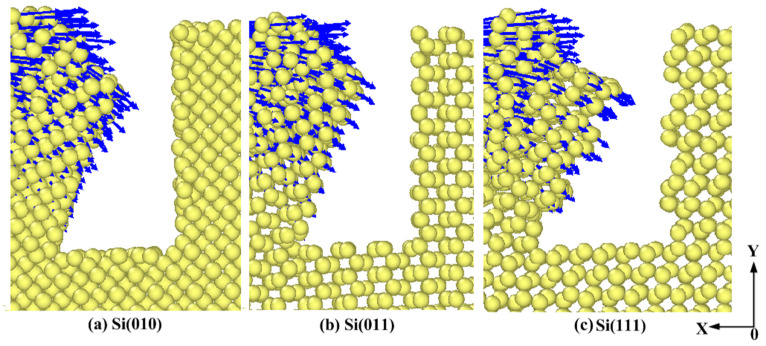
Effects of anisotropy on the deformation of the first pore at the polishing distance of 4 nm.

**Figure 6 micromachines-11-00742-f006:**
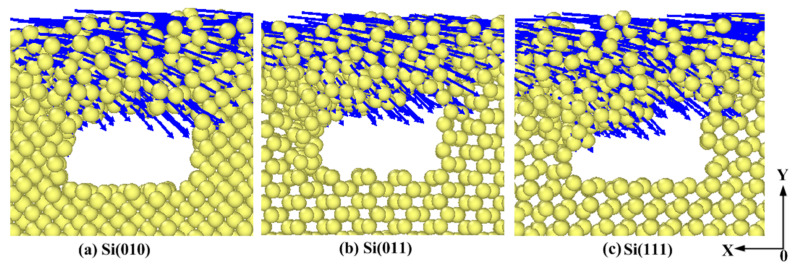
Effects of anisotropy on the deformation of the first pore at the polishing distance of 14.2 nm.

**Figure 7 micromachines-11-00742-f007:**
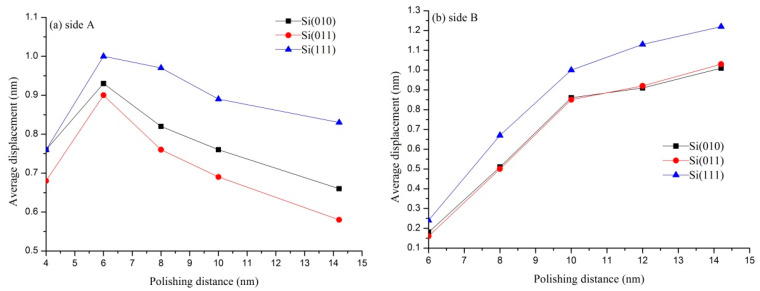
Average displacement of A and B.

**Figure 8 micromachines-11-00742-f008:**
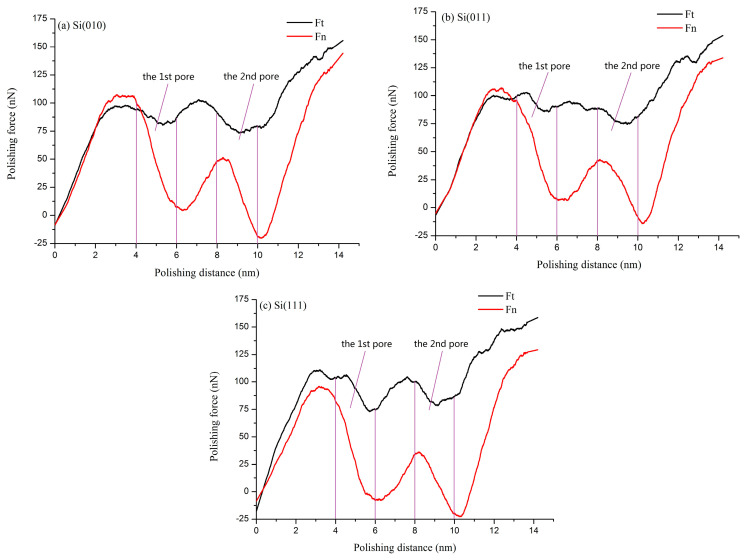
Effects of anisotropy on the polishing force at the polishing depth of 1.0 nm.

**Figure 9 micromachines-11-00742-f009:**
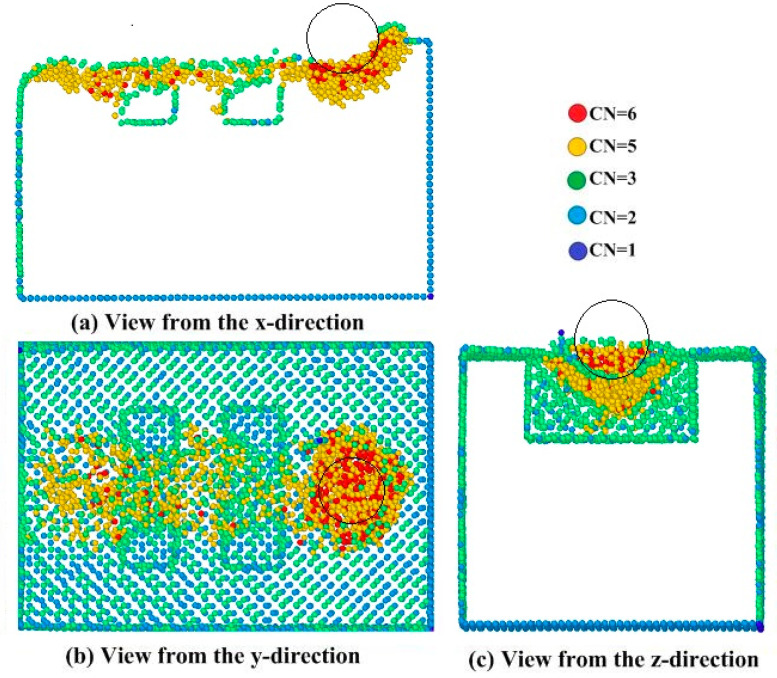
A diagram of the cross-sectional views of the atomic positions with various views at the polishing distance of 14.2 nm. CN, coordination number.

**Figure 10 micromachines-11-00742-f010:**
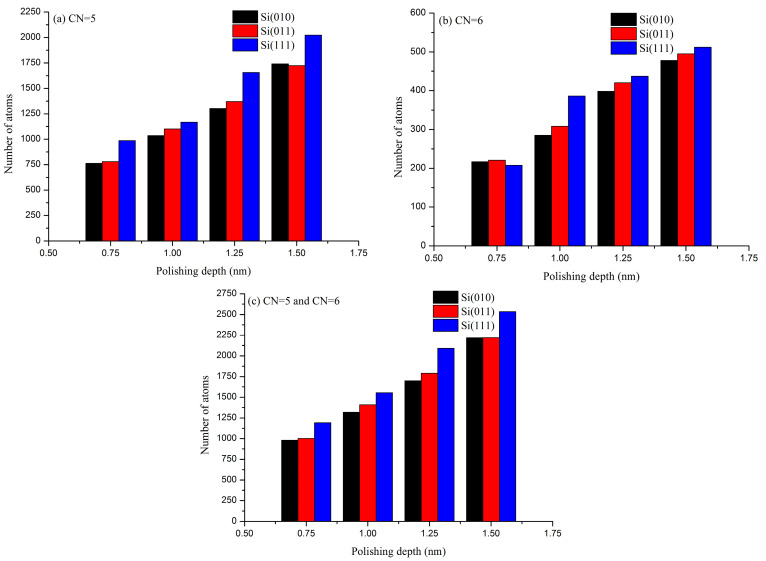
Effects of anisotropy on the number of atoms with coordination numbers 5 and 6 at different polishing depths.

**Figure 11 micromachines-11-00742-f011:**
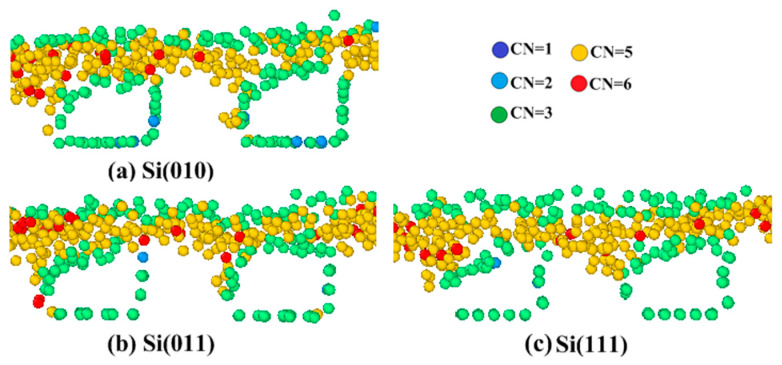
Effects of anisotropy on pore walls at polishing depth of 1.0 nm.

**Table 1 micromachines-11-00742-t001:** MD (molecular dynamics) simulation parameters in 3D nano-machining.

Parameters	Value
Radius of abrasive	2 nm
Atom number of abrasive	5899
Size of workpiece	17 × 11 × 11 nm^3^
Atom number of workpiece	Si(010): 102241 Si(011): 102374 Si(010): 101493
Time step	1 fs
Initial temperature	293 K
Polishing depth (ap)	0.75 nm, 1 nm, 1.25 nm, 1.5 nm
Polishing distance	14.2 nm
Polishing direction	[-100] on (010) surface [-100] on (011) surface [-11-2] on (111) surface
Size of pore	2 × 3 × 6 nm^3^
Polishing speed	100 m/s
Spinning speed	25 m/s

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
