# Peer review of "Effects of Anisotropy on Single Crystal Silicon in Polishing Non-Continuous Surface"

_micromachines, 2020, doi:10.3390/mi11080742_

Round 1

Reviewer 1 Report

Dear Authors,

Please find attached the comments related to the submitted manuscript.

Author Response

Responses to Reviewer 1 Comments

Comments: The submitted manuscript presents molecular dynamics simulation results for non-continuous surface of a single crystal silicon structure when subjected to a diamond abrasive polishing process. The flow of polishing process and its effects on the polished single crystal silicon structure is analysed and characterized for different orientations (i.e. three surfaces are considered in this study, Si (010), Si(011) and Si (111)) by considering surface morphology, displacement, polishing force and phase transformation. A high degree of similarity was found between this paper and a previously (but very recently) published paper, by common authors, i.e. reference 30, in the submitted manuscript (G. Wang et al, Materials Science in Semiconductor Processing 2020, 118, 105168). In this submitted paper, a similar method and system is presented (as that reported in reference 30), excepting the fact that here are discussed effects like surface morphology, atomic displacement, polishing force and phase transformation for all three surfaces, as compared to (010) surface presented alone in reference 30. The overall conclusions presented in both papers are very similar. Unless the authors would be able to clearly explain and highlight the significant advance in this field, that would be brought by publishing this paper, as compared to other previously submitted papers, I would recommend rejection of this paper.

Response: Thanks for the reviewer’s criticism and suggestions on this paper. In response to the issues involved in the reviewer’s comments, I would like to elaborate on the innovations of this paper and the progress of research in this area.

First of all, this paper mainly explores how the crystal structure of materials (crystallographic orientations) influences the polishing process of the pore structure. The previous paper mainly focuses on the influence of the geometric structure in the workpiece and the difference between the porous surface and continuous surface in the polishing process. It does not involve the structure of the crystal itself.

Secondly, this paper proposes the influence of the crystallographic orientations on the deformation mechanism and removal mechanism of materials at pore walls: the deformation of the A side in the Si(010) and Si(011) workpieces belongs to the overall deformation, while the A side of the Si(111) workpiece belongs to the local deformation. In the whole polishing process, the removal of materials at pore walls in the Si(111) workpiece is the largest, which is related to the cleavage characteristics of the single crystal silicon.

In addition, laws that the polishing force is affected by crystal orientations in the process of polishing porous materials have also been proposed: for the tangential force, crystallographic orientations mainly affects the position where the tangential force reaches the maximum and minimum values, and the drop of the tangential force is also affected by crystallographic orientations, which is related to the accumulation of chips in different crystal orientations. For the normal force, the crystal orientation has little effect on it.

Finally, the phase transformation of the workpiece during the polishing process and deterioration laws of the pore walls in different crystal orientations were studied. This study found that atoms with a coordination number of 5 in the Si(111) workpiece are always the most. For atoms with a coordination number of 6, the influence of the crystal orientation on it is basically the same as that of atoms with a coordination number of 5. There are only some small local differences. Among three types of crystal surfaces, the total number of main phase transformation atoms on the Si (111) surface is the largest, and the total number of main phase transformation atoms on the Si (010) surface is the smallest. The deterioration of the pore wall is mainly reflected in the overall deterioration of pore walls in the Si(010) and SI(011) workpieces, while the Si(111) workpiece only has the partial deterioration of the pore wall.

These rules are all put forward for the first time in this paper.

Point 1: The molecular dynamics simulation model could be better described. Some relevant equations could be added in the manuscript. The software used for performing molecular dynamics simulations should be mentioned. The total particles number used in the simulation should be mentioned.

Response 1: Related formulas (equations about the potential functions) have been added to describe the modeling process. All MD simulations are based on the large-scale Atomic/Molecular Massively Parallel Simulator (LAMMPS) and results are visualized by OVITO. The number of atoms in the abrasive and workpiece in the simulation process is described in the article. The total number of atoms in the abrasive is 5899, and the total number of atoms in Si(010), Si(011) and Si(111) workpieces are 102241, 102374 and 101493, respectively. (revised at lines 148-172, 176-178 and 181 on pages 4-6)

Point 2: How the authors choose the sizes of the pores? From which considerations?

Response 2: The selection of pores mainly refers to the following aspects: 1 The size of the workpiece surface and the size of the abrasive (The X-direction width of the pore is considered to be smaller than the diameter of the abrasive and the size of the pore is smaller than the length of each side of the surface.) 2 The pore occupies the surface space of the workpiece should be reasonable (The X-direction of the pore needs to leave enough continuous surface polishing distance, and the Z-direction should not be too close to the model boundary.) 3 The distance between the two pores should be reasonable (The main point is that the X-direction distance should not be too short. If it is too short, the two pores cannot be distinguished well, and the pore wall is likely to be too thin.)

Point 3: Page 4, figure 2, the red and blue circles should be added in the figure legend (should be described).

Response 3: The necessary description has been added to the different colored atoms in Figure 2. Red atoms are atoms on the face of the unit cell and the blue atoms are atoms in the unit cell. (revised at lines 139-140 on page 4)

Point 4: Always use 293 K instead of 293K or 1.5 nm instead of 1.5nm (example, page 6, line 175).

Response 4: All units in the paper have been adjusted to add spaces.

Point 5: Page 5, table, 293 K for initial temperature should be corrected.

Response 5: The data has been corrected in the table 1. (revised at line 181 on page 6)

Point 6: Figure 4 Arrows indicating chips formation or side flows could be added. Some color legend should added.

Response 6: Arrows have been added in the Figure 4. (revised at lines 206-208 on page 7)

Point 7: Figure 4b, ap=1 nm, is this picture correct? It is placed in proper spot?

Response 7: The picture here has been carefully checked and re-uploaded for correction. (revised at lines 206-208 on page 7)

Point 8: Figure 5, 6 axes could be added.

Response 8: Axes have been added in Figure 5, 6. (revised at lines 238-242 on page 8)

Point 9: Page 7, line 211, a lower text font (0.5×3×4) should be used.

Response 9: A lower text font (0.5×3×4) has been used. (revised at line 245 on page 8)

Point 10: Page 10, line 339 polishing depth of 14.2 nm instead of 1 nm?

Response 10: 14.2 nm is the polishing distance, 1 nm is the polishing depth, which has been explained in the paper (lines 370-371 and 377-378 at pages 11-12).

Point 11: Page 10, line 339, …”a large number of under abrasive…” a word is missing here…

Response 11: The missing word has been added. (revised at line 371 on page 11)

Point 12: Page 11, Figure 9, define CN.

Response 12: CN is the coordination number, which has been defined in the paper. (revised at line 375 on page 11)

Point 13: Page 11, line 359 …”and the difference is small” could be deleted.

Response 13: "and the difference is small" has been deleted.

Point 14: Page 11, line 369 and 370, verify and correct the values

Response 14: The data in the paper has been verified and corrected. (lines 394-395 on page 12)

Point 15: Verify and correct the author names listed in the references (see for example Ref 2, 3, 16, 19 30 and others as well).

Response 15: The author names listed in the references have been verified and corrected.

Author Response

Responses to Reviewer 2 Comments

Comments: The presented manuscript is devoted to an interesting topic, it is well written and well organized. Results of molecular dynamics simulations are the additional advantage of the manuscript. The manuscript can be accepted for publication after minor corrections listed below.

  1. Please, enlarge the Introduction part a little and refer to the experimental/theoretical works, where the comparison of experiment and molecular dynamics simulations is discussed. I recommend paying more attention to the works [1 - 2].

References

[1] Superhard CrN/MoN coatings with multilayer architecture

Materials and Design153, pp. 47-59, 2018

[2] Experimental and theoretical studies of the physicochemical and mechanical properties of multi-layered TiN/SiC films: Temperature effects on the nanocomposite structure

Composites Part B: Engineering, 142, pp. 85-94,2018

Response: In response to the questions raised by the reviewer, I added some additional descriptions to the introduction of the paper and referred to the experimental research results in above references. (revised at lines 76-80 and 504-510 on pages 2 and 15)

Reviewer 3 Report

Authors present the classical MD investigation of the polishing process of the single crystal silicon. I guess the following questions should be clarified before publication:
1)Authors pointed in line 137 that the microcanonical ensemble (NVE) with constant energy of system is used. How it posssibe, if in part of the system the fixed temperature is kept? Moreover, the external force should be applied to move abrasive with constant speed.
2)Which is the final state of the system (structures in Fig. 6)? Initially open pores becomes close after the polishing or not?
3)How the obtained values of the polishing forces relate to the experimental ones?
4)Why the polishing speed is taken equal to 100 m/s?
5)Which is the temperature in the layer contacted with abrasive? This temperature exceed melting point or not?

Author Response

Responses to Reviewer 3 Comments

Authors present the classical MD investigation of the polishing process of the single crystal silicon. I guess the following questions should be clarified before publication:

Point 1: Authors pointed in line 137 that the microcanonical ensemble (NVE) with constant energy of system is used. How it posssibe, if in part of the system the fixed temperature is kept? Moreover, the external force should be applied to move abrasive with constant speed.

Response 1: Although the micro-canonical ensemble is adopted in this paper, the temperature of the specified local area can be adjusted by the existing LAMMPS program to meet the conditions required for simulations.

In this paper, a constant feed speed and a rotation speed are set in the initial conditions of the simulation, and the speed does not change during the polishing process, which is the same as adding force to maintain a constant speed.

Point 2: Which is the final state of the system (structures in Fig. 6)? Initially open pores becomes close after the polishing or not?

Response 2: The three pictures in Fig. 6 all show the final state of the 1st pore at the end of polishing. The final state of the system is at the polishing distance of 14.2 nm.

At the end of polishing, the pores are partially covered.

Point 3: How the obtained values of the polishing forces relate to the experimental ones?

Response 3: We referred to the experimental and simulation data and the fluctuation law of the cutting force in the existing papers. Besides, we compared it with the laws of the polishing forces obtained in this paper, and judged the rationality of the polishing forces.

Point 4: Why the polishing speed is taken equal to 100 m/s?

Response 4: Existing molecular dynamics simulation studies have shown that the simulation results obtained by cutting simulations with speeds between 0-200 m/s are reliable. But if the speed is set to be relatively small, it will consume a lot of computing time and occupy computing resources. Therefore, in order to improve the simulation efficiency, the polishing speed is set to 100 m/s under the premise of ensuring reasonableness.

Point 5: Which is the temperature in the layer contacted with abrasive? This temperature exceed melting point or not?

Response 5: This temperature is the average temperature of the Newtonian layer.

According to the data I have simulated, the average temperature of the Newtonian layer does not exceed 380 K at the maximum cutting depth, which is far below the melting point of single crystal silicon.

Reviewer 4 Report

The paper describes a molecular dynamic study of machining an interrupted surface at the nanoscale. However, there is no experimental detail to support the analysis. The paper should be revised to include experimental proof. A spell check is required for this paper.

Author Response

Responses to Reviewer 4 Comments

Comments: The paper describes a molecular dynamic study of machining an interrupted surface at the nanoscale. However, there is no experimental detail to support the analysis. The paper should be revised to include experimental proof. A spell check is required for this paper.

Response: Thanks for the reviewer’s valuable comments, molecular dynamics has caused certain difficulties in the experimental verification due to its small scale, and placed extremely high requirements on experimental equipment. However, the formulas describing atomic interaction in molecular dynamics have been widely used in molecular dynamics simulation research, and some scholars have conducted experimental research and found that the corresponding potential function is used to calculate the atomic interaction in a specific simulation environment, which can accurately describe the cutting and polishing process. The Morse potential and Tersoff potential used in this paper have been widely used in the molecular dynamics simulation process of diamond cutting single crystal silicon, which is consistent with the existing experimental results, and there are a large number of documents that can prove their rationality. For example, the reference[ 31] gives a detailed description of the Tersoff potential function.

Reviewer 5 Report

This is an interesting study as the authors attempted to examine the non-continuous single crystal silicon surfaces from the aspects of surface morphology, displacement, polishing force and phase transformation. They have shown that the Si(010) surface is easier to accumulate chips than the Si(011) surface and the Si(111) surface. They examined the anisotropy effect on the pore walls, using a molecular dynamics model. The basic science and methodologies discussed are correct to a large extent. However, I have a few suggestions that the authors must consider to improve their ms. The writing quality throughout the ms is somehow poor. This enforced me to understand what the authors wanted to say at times, starting from the second line of the abstract. Accordingly, careful discussion and rewriting various other parts of the paper are essential. The authors should consult a native English writer to help them, if this is impossible for them to correct by themselves.

  • Fonts in Fig. 1 should be enlarged. It is no good in its current form.
  • “Although above researches have conducted many studies on the anisotropy …” does not read good. Wring improvement is necessary.
  • Which software was used to generate Figs. 7, 8, 10? This should be cited. High resolution picture should be provided.
  • The conclusion part of the paper is too long. It should be reduced to a maximum of two paragraphs, with a total of 20-25 lines, highlighting the essence of the finding.

Author Response

Responses to Reviewer 5 Comments

Comments: This is an interesting study as the authors attempted to examine the non-continuous single crystal silicon surfaces from the aspects of surface morphology, displacement, polishing force and phase transformation. They have shown that the Si(010) surface is easier to accumulate chips than the Si(011) surface and the Si(111) surface. They examined the anisotropy effect on the pore walls, using a molecular dynamics model. The basic science and methodologies discussed are correct to a large extent. However, I have a few suggestions that the authors must consider to improve their ms. The writing quality throughout the ms is somehow poor. This enforced me to understand what the authors wanted to say at times, starting from the second line of the abstract. Accordingly, careful discussion and rewriting various other parts of the paper are essential. The authors should consult a native English writer to help them, if this is impossible for them to correct by themselves.

Response: Thanks for the reviewer’s valuable comments, the English of this paper has been improved.

Point 1: Fonts in Fig. 1 should be enlarged. It is no good in its current form

Response 1: The Fonts in Figure 1 has been adjusted, and the font size has been appropriately increased to make its proportions more appropriate. (revised at lines 125-126 on page 3)

Point 2: “Although above researches have conducted many studies on the anisotropy …” does not read good. Wring improvement is necessary.

Response 2: The sentence in the paper has been revised. (revised at lines 111-112 on page 3)

Point 3: Which software was used to generate Figs. 7, 8, 10? This should be cited. High resolution picture should be provided.

Response 3: These pictures are processed by the ORIGIN, which has been explained in the paper, and higher resolution pictures have been provided. (revised at line 178 on page 5)

Point 4: The conclusion part of the paper is too long. It should be reduced to a maximum of two paragraphs, with a total of 20-25 lines, highlighting the essence of the finding.

Response 4: The conclusion part has been streamlined and optimized. The conclusion is controlled within 25 lines. However, the three conclusions mentioned in the paper belong to three different types of analysis. I think three paragraphs can provide more logical thinking than two paragraphs. (revised at lines 435-459 on page 14)

Round 2

Reviewer 1 Report

Dear Authors,

The reviewer comments and suggestions were properly addressed. The revised manuscript is found now suitable to be published in Micromachines MDPI Journal after minor revisions.
As previously mentioned, in the first review, the authors list from certain references is not properly written. Please recheck the author names for references 2, 3, 16, 19 and 30 (in the first submitted manuscript).
As an example, for reference 16 the name of the author was changed by its surname, i.e. "David Bürger" should be listed as Bürger D. et. al, instead of "David B. et. al"
Also, for example in reference 2, "Luo XC" should appear as "Luo X.", the same for all authors. (only the first letter of the surname should be mentioned).

Author Response

Comments: The reviewer comments and suggestions were properly addressed. The revised manuscript is found now suitable to be published in Micromachines MDPI Journal after minor revisions.

As previously mentioned, in the first review, the authors list from certain references is not properly written. Please recheck the author names for references 2, 3, 16, 19 and 30 (in the first submitted manuscript).

As an example, for reference 16 the name of the author was changed by its surname, i.e. "David Bürger" should be listed as Bürger D. et. al, instead of "David B. et. al"

Also, for example in reference 2, "Luo XC" should appear as "Luo X.", the same for all authors. (only the first letter of the surname should be mentioned).

Responses: All references have been carefully checked and revised as required.

Reviewer 4 Report

Please include experimental results to confirm your analyses, not the analysis of other researchers.

Author Response

Comments: Please include experimental results to confirm your analyses, not the analysis of other researchers.

Responses: Thanks for the reviewer’s comments. However, the simulation scale of this paper has reached the nanoscale, so it is difficult to conduct experiments. At the same time, molecular dynamics is a scientific simulation method, which is often used in nanoscale and other scales that are not convenient for experiments to reveal the processing mechanism. In fact, there are quite a few papers using molecular dynamics simulation because the scale is too small to carry out experiments. We sincerely hope that you can understand our difficulties.

Reviewer 5 Report

The authors of the study have considered all of my comments made in my previous review. They have significantly modified the text, and the paper is now understandable to some large extent. The pictures are modified. Discussion of basic science is also improved, together with the conclusion part of the paper. While this paper may be suitable for publication, I still ask the authors to seriously revise the paper (grammar and typos!). In fact, there are still many sentences in the ms that are not understandable. A few  such examples could be:

1- The authors wrote on Page 5 that ... Although these researches have carried out many studies, the anisotropy of porous crystalline materials still needs to be described in detail...

Concern: Which research does what studies?

2- All MD simulations are based on the large-scale Atomic/Molecular Massively Parallel Simulator (LAMMPS)and results are visualized by OVITO.Curvesaredrawn by ORIGIN.

Why is there no citation to Origin, Ovito, and LAMMPS?

3- The conclusion section is not reduced as suggested.

And there are several phrases and sentences throughout the ms that must be seriously revised.

Author Response

Comments: The authors of the study have considered all of my comments made in my previous review. They have significantly modified the text, and the paper is now understandable to some large extent. The pictures are modified. Discussion of basic science is also improved, together with the conclusion part of the paper. While this paper may be suitable for publication, I still ask the authors to seriously revise the paper (grammar and typos!). In fact, there are still many sentences in the ms that are not understandable. A few such examples could be:

Responses: Thanks for the reviewer’s comments. The grammar and typos in the paper have been checked in detail and corrected.

Point 1: The authors wrote on Page 5 that ... Although these researches havecarried out many studies, the anisotropy of porous crystalline materials still needs to be described in detail...

Concern: Which research does what studies?

Responses 1: The research mentioned in the paper is about the anisotropy of non-porous crystalline materials, and the corresponding description in the paper has been revised. (revised at lines 109-110 on page 3)

Point 2: All MD simulations arebased on the large-scale Atomic/Molecular Massively Parallel Simulator (LAMMPS)and results are visualized by OVITO. Curves are drawn by ORIGIN.

Why is there no citation to Origin, Ovito, and LAMMPS?

Responses 2: The necessary references have been added to the paper. (revised at lines 175-177 and 525-529 on page 5 and 16)

Point 3: The conclusion section is not reduced as suggested. And there are several phrases and sentences throughout the ms that must be seriously revised.

Responses 3: The conclusions have been further refined, shortened into two paragraphs with 22 lines. Meanwhile, some sentences in the paper are revised and corrected. (revised at lines 433-451 on page 14)
